# Hyperbolic Residual Quantization: Discrete Representations for Data with Latent Hierarchies

## Abstract

Hierarchical data arise in countless domains, from biological taxonomies and organizational charts to legal codes and knowledge graphs. Residual Quantization (RQ) is widely used to generate discrete, multitoken representations for such data by iteratively quantizing residuals in a multilevel codebook. However, its reliance on Euclidean geometry can introduce fundamental mismatches that hinder modeling of hierarchical branching, necessary for faithful representation of hierarchical data. In this work, we propose Hyperbolic Residual Quantization (HRQ), which embeds data natively in a hyperbolic manifold and performs residual quantization using hyperbolic operations and distance metrics. By adapting the embedding network, residual computation, and distance metric to hyperbolic geometry, HRQ imparts an inductive bias that aligns naturally with hierarchical branching. We claim that HRQ in comparison to RQ can generate more useful for downstream tasks discrete hierarchical representations for data with latent hierarchies. We evaluate HRQ on two tasks: supervised hierarchy modeling using WordNet hypernym trees, where the model is supervised to learn the latent hierarchy - and hierarchy discovery, where, while latent hierarchy exists in the data, the model is not directly trained or evaluated on a task related to the hierarchy. Across both scenarios, HRQ hierarchical tokens yield better performance on downstream tasks compared to Euclidean RQ with gains of up to $20\%$ for the hierarchy modeling task. Our results demonstrate that integrating hyperbolic geometry into discrete representation learning substantially enhances the ability to capture latent hierarchies.

## 1 Introduction

Hierarchical structures appear throughout human knowledge and information organization, serving as essential frameworks for understanding complex relationships between entities. These structures can be found in biological classifications of living organisms (Mayr, 1968), business organizational structures (Chandler Jr, 1969), and computer file systems (McKusick et al., 1984). Studies show that when children learn, they organize their knowledge in hierarchies (Inhelder and Piaget, 2013). This pattern extends to numerous other domains as well: from taxonomic categorization in libraries and archives to the nested organization of legal codes and regulations. Government systems typically follow hierarchical arrangements, with federal, state, and local levels each containing their own internal hierarchies. Similarly, academic disciplines are organized into fields, subfields,

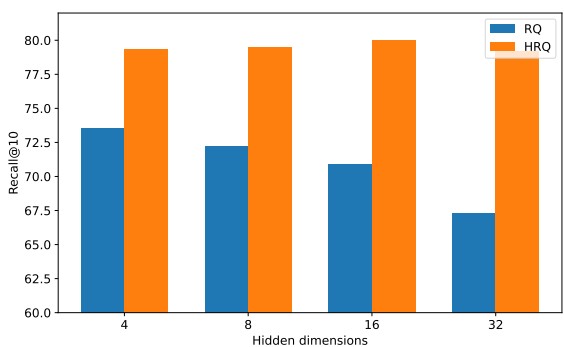

Figure 1: Recall@10 of the hypernym generation based on tokens generated by HRQ vs tokens generated by RQ. HRQ consistently outperforms RQ. Furthermore, HRQ sustains consistent scores across different dimensionalities of the embedding.

and specialized areas. The presence of hierarchical structures across such diverse domains reflects their importance in how humans conceptualize and organize information.

In the current state of machine learning modeling, most often continuous vectors are used to represent entities (Bengio et al., 2003; Mikolov, 2013; Bordes et al., 2013). However, it can sometimes be beneficial to use discrete representations rather than continuous vector embeddings. Discrete tokens function effectively as labels because in the discrete domain, generation is equivalent to prediction. This equivalence allows models to avoid complicated generation methods like GANs (Goodfellow et al., 2020) or diffusion models (Ho et al., 2020), and instead rely on more straightforward prediction tasks. Discrete representations also tend to be more interpretable as each token can correspond to a specific concept or attribute in the hierarchy (Rajput et al., 2023).

Residual Quantization Variational Autoencoders (RQ-VAE) (Lee et al., 2022; Zeghidour et al., 2021) leverage these benefits by creating semantic hierarchical discrete representations through a multilevel quantization process (van den Oord et al., 2017). At each step of residual quantization, the model encodes increasingly fine-grained details, with earlier levels capturing broader structural elements and later levels representing more specific attributes. The result is a list of tokens that together create an identifier of the entity. We will refer to this hierarchical discrete representation as *Multitoken(MT)*. By learning discrete tokens at multiple levels of abstraction, RQ-VAE provides a framework for modeling hierarchical relations directly from dense embedding.

However, RQ-VAE operates within Euclidean space, which imposes fundamental limitations on its ability to capture hierarchical relationships. Euclidean geometry struggles to efficiently represent tree-like structures (Gromov, 1987), as the volume of space grows polynomially with distance from the origin, while the number of nodes in a hierarchy typically grows exponentially with depth. This geometric mismatch means that Euclidean-based models like RQ-VAE inevitably lose important hierarchical information during encoding.

In contrast, hyperbolic space (Gromov, 1987), a Riemannian manifold with constant negative curvature, has been shown to model hierarchies remarkably well (Nickel and Kiela, 2017; 2018). The hyperbolic space can approximately isometrically embed any tree already in two dimensions (Gromov, 1987), whereas the same cannot be said for the Euclidean space of any dimension. The volume in the hyperbolic space grows exponentially with distance from the origin, aligning well with the growth of number of nodes in hierarchy.

The ability to encode trees by hyperbolic geometry has inspired numerous advances in machine learning. Hyperbolic neural networks have also been extensively leveraged in continuous embedding models to exploit latent hierarchies in a variety of domains. Poincaré embeddings (Nickel and Kiela, 2017), learn continuous hierarchies by mapping symbolic data into an n-dimensional Poincaré ball. The authors showed that these embeddings outperform the Euclidean ones on tree-structured data in terms of both representation capacity and generalization ability. Hyperbolic embeddings found use in data domains rich in latent hierarchies, like knowledge-graph representation(Balazevic et al., 2019; Chami et al., 2020b; Liang et al., 2024) and recommender systems (Sun et al., 2021; Chamberlain et al., 2019; Mirvakhabova et al., 2020), and other (Wilson, 2021; Ganea et al., 2018c).

Despite these advances in continuous embedding models, the application of hyperbolic geometry to discrete representation learning has remained mostly underexplored. HyperVQ (Goswami et al., 2024) proposes to perform vector quantization in a hyperbolic space by phrasing it as a hyperbolic multinomial logistic regression. In this paper, we use hyperbolic distance to find the nearest codebook vector and focus on the hyperbolic version of Residual Quantization. We introduce Hyperbolic Residual Quantization (HRQ), which performs residual quantization (RQ) in a hyperbolic space with an adapted process of residual quantization to accommodate the hyperbolic structure. We claim that for **data with latent hierarchies** residual quantization benefits from hierarchical inductive bias induced by hyperbolic space. We implement this approach through several key adaptations: first, we employ hyperbolic neural networks for the embedding process, ensuring that data representations reside natively in hyperbolic space. Second, we utilize hyperbolic operations to calculate the residuals between quantization levels, preserving the geometric properties of the space throughout the quantization process. Finally, we incorporate hyperbolic distance metrics in the clustering algorithm, allowing the model to properly capture the hierarchical relationships between data points. These modifications enable HRQ-VAE to make use of the natural advantages of hyperbolic geometry to represent hierarchical structures while maintaining the benefits of discrete token-based representations.

We evaluate the quality of the multitokens created by HRQ in two scenarios. First, we test its ability to model hierarchies with supervision on the hierarchy. (H)RQ creates multitokens of nouns (Miller, 1995) based on their hypernymy relation. Then, we test which representation is more useful in generating the hypernym for a given noun. We show that multitokens learned with Hyperbolic Residual Quantization significantly outperform tokens learned with Residual Quantization. Furthermore, we test the model's ability to create meaningful hierarchies without direct supervision on the hierarchy. Specifically, we evaluate it in a scenario where hierarchy exists, but the model is not supervised on modeling the hierarchy and is used for a task not directly related to the hierarchy. We show that the multitokens generated by HRQ outperform the multitokens generated by RQ in this scenario as well.

The paper is organized as follows. In Section 2.1 we introduce necessary concepts from the theory of hyperbolic spaces for our method and describe the RQ-VAE algorithm. In Section 3 we describe HRQ-VAE. In Section 4 we demonstrate our experimental results. In section 5. Finally, in Section 6 we summarize our findings and propose future directions.

## 2 BACKGROUND

### 2.1 HYPERBOLIC SPACE

Hyperbolic geometry operates on manifolds with constant negative Gaussian curvature. A fundamental characteristic of hyperbolic geometry is its exponential spatial expansion relative to the distance from any reference point, creating abundant capacity to represent branching structures. This property enables hyperbolic spaces to accommodate the embedding of complex hierarchical relationships with minimal distortion. Research has demonstrated that arbitrary tree structures can be embedded within a hyperbolic space while approximately preserving their metric properties (Gromov, 1987; Hamann, 2018). Because of these results, hyperbolic space can be conceptualized as "a continuous version of a tree," making it exceptionally valuable for computational representations of hierarchical data structures, complex networks with inherent branching patterns, and systems characterized by nested relationships.

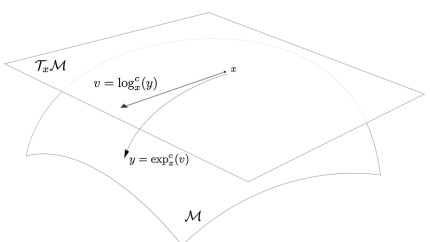

Figure 2: Visualization of the tangent space and related operations. Exponential map $exp_x^c$ maps from the tangent space attached at $x$ to the manifold and logarithmic map $log_x^c$ maps from the manifold to the tangent space attached at point $x$.

In this work, we use the Poincaré ball model, which is the most widely used representation of the hyperbolic space in the context of neural networks. The definition of the Poincaré ball we use follows Ganea et al. (2018a).

**The Poincaré Ball Model.** The $n$-dimensional Poincaré Ball $\mathbb{P}_c^n$ with curvature $c$ is a set $\{x \in \mathbb{R}^n : c||x||^2 < 1\}$ with Riemannian metric $g_x^{\mathbb{P}} = \lambda_x^2 g^E$, where $g^E$ is the Euclidean metric tensor and $\lambda_x := \frac{2}{1-c||x||^2}$. The gyrovector spaces (Ungar, 2008) allow one to define the operations corresponding to the standard operations in the euclidean vector spaces. In the Poincaré ball model $\mathbb{P}_c^n$ the **Möbius addition** is a hyperbolic analogue of a standard addition operation, defined as

$$x \oplus_c y := \frac{(1 + 2c\langle x, y \rangle + c||y||^2)x + (1 - c||x||^2)y}{1 + 2c\langle x, y \rangle + c^2||x||^2||y||^2} \quad \Big| \quad x \ominus_c y := x \oplus_c (-y)$$

**Hyperbolic Distance.** Distance in the Poincaré ball model of the hyperbolic space is defined as

$$d^{\mathbb{P}_c}(u, v) = \text{arcosh}(1 + 2\frac{c||u - v||^2}{(1 - c||u||^2)(1 - c||v||^2)}) \tag{1}$$

As points get farther from the center, the distance between them grows exponentially, creating increasingly more space near the boundaries. Conversely, there is limited space near the center, naturally constraining which points can occupy these central positions - a property that aligns with hierarchical structures where few elements serve as high-level abstractions. The metric treats distance

differently when moving toward/away from the center versus moving side-to-side, which helps capture both how deep items are in the hierarchy and how they branch apart. Items that belong to the same branch end up close to each other but at different depths, while the exponential growth of distances ensures effective separation between different branches. This makes it easy to preserve both the local structure (items close to each other in the hierarchy) and the overall organization (how different branches relate to each other).

**The Tangent Space.** The tangent space $\mathcal{T}_x\mathcal{M}$ of the manifold $\mathcal{M}$ at point $x$ is an euclidean space attached to the manifold at point $x$ that intuitively contains all possible velocities the vector attached to $x$ can have.

In order to translate between manifold and tangent space, two special maps are used. The exponential map projects vectors from the tangent space $\mathcal{T}_x\mathcal{M}$ to the manifold $\mathcal{M}$. In contrast, the logarithmic map is used to project from the manifold $\mathcal{M}$ to the tangent space $\mathcal{T}_x\mathcal{M}$. For the Poincaré ball model, the exponential and logarithmic maps are equal to

$$\exp_x^c(v) = x \oplus_c \left( \tanh\left( \sqrt{c}\frac{\lambda_x \|v\|}{2} \right) \frac{v}{\sqrt{c}\|v\|} \right)$$

$$\log_x^c(y) = \frac{2}{\sqrt{c}\lambda_x} \tanh^{-1}(\sqrt{c}\| - x \oplus_c y\|)\frac{-x \oplus_c y}{\| - x \oplus_c y\|}$$

Similarly to the addition, scalar multiplication has its own hyperbolic version. These operations suffice to derive linear layers. Furthermore, with exponential and logarithmic maps, it is possible to add nonlinearities by translating back and forth from the manifold to the tangent space. Here, we defined only necessary the concepts that will be explicitly used in the HRQ-VAE algorithm, and omitted others that are necessary to derive hyperbolic layers (like scalar multiplication). We refer interested readers to Ganea et al. (2018a) or Cannon et al. (1997)

## 3 METHOD

**Multitoken(MT)** is a list of discrete tokens that *together* identify an entity in the dataset $D$. While the typical flat discrete representation is a single number from 0 to $|D| - 1$, the multitoken of length $k$, is a list of $k$ tokens $[t_0, ..., t_{k-1}]$, such that jointly they identify a corresponding entity. If multitokens are structured in a semantic way, they can offer representational benefits over flat tokens. Specfically, tokens can be shared across different multitokens, leading to information sharing and a more efficient and robust representation than flat tokens, where each entity is treated independently. For example, a tiger might be identified by a multitoken [12,24] and a lion might be identified by a token [12,364]. In this case, the first token 12 is shared between the two entities and leads to a shared part of the representation. The difficulty lies in creating good, structurally semantic multitokens.

### 3.1 HYPERBOLIC RESIDUAL QUANTIZATION.

**Hyperbolic Residual Quantization (HRQ)** is a method for hierarchical multitoken representation that performs residual quantization directly in hyperbolic space. The method is inspired by classical residual quantization (RQ), which approximates vectors by iteratively quantizing their residuals with respect to multiple codebooks. HRQ generalizes this process to hyperbolic geometry, ensuring that the hierarchical structure induced by quantization is better aligned with latent hierarchies in the data.

Let $C = [C_0, \ldots, C_{k-1}]$ be a sequence of codebooks, where each $C_i$ contains $s$ vectors in $\mathbb{P}_c^h$, the $h$-dimensional Poincaré ball of curvature $c$. For a vector $x_s^{\mathbb{P}_c} \in \mathbb{P}_c^h$, HRQ produces a sequence of tokens $[t^0, t^1, \ldots, t^{k-1}]$ and corresponding codebook embeddings $[e^0, e^1, \ldots, e^{k-1}]$ as follows. The initial residual is set to $r^0 = x_s^{\mathbb{P}_c}$. At each step $i$, we quantize the current residual using hyperbolic distance:

$$e^i, t^i = q_{C_i}^{\mathbb{P}_c}(r^i), \quad \text{where } e^i \in C_i, \ t^i \in \{0, \ldots, s-1\}.$$

The residual is then updated via Möbius subtraction, $r^{i+1} = r^i \ominus_c e^i$, and the process repeats until $k$ tokens are obtained. The multitoken $[t^0, t^1, \ldots, t^{k-1}]$ uniquely identifies the representation of $x_s^{\mathbb{P}_c}$. The reconstruction is given by the Möbius sum of selected embeddings denoted as $y_s^{\mathbb{P}_c} = \bigoplus_{i=0}^{k-1} e^i$.

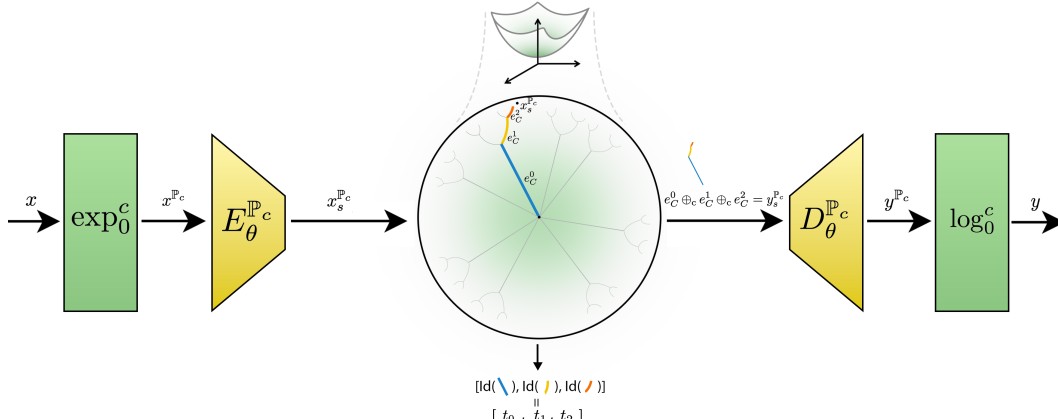

Figure 3: HRQ-VAE visualized. In the image HRQ-VAE quantizes given vector $x$ into a multitoken $[t_0, t_1, t_2]$ and its corresponding embeddings $e_C^0, e_C^1, e_C^2$. Green blocks represent mapping to and from hyperbolic space. Yellow blocks represent hyperbolic autoencoder. The detailed part in the middle is responsible for hyperbolic residual quantization. The space expands exponentially the further we go away from the center. In fact, The circle's border is at infinite distance from point 0. As a consequence, most of the points must be distant from the center and only a small number of points can be at a privileged position close to the center. This leads to natural occurence of hierarchies. Light gray branches represent the possible HRQ-VAE and

We denote the quantization process by $HRQ_C(x_s^{\mathbb{P}_c}) = \big([t^0, \ldots, t^{k-1}], y_s^{\mathbb{P}_c}\big)$. In cases where the final residual matches the last codebook, i.e. $r^{k-1} \in C_{k-1}$, the reconstruction is exact, $y_s^{\mathbb{P}_c} = x_s^{\mathbb{P}_c}$. Otherwise, the approximation error $d_{\mathbb{P}_c}(x_s^{\mathbb{P}_c}, y_s^{\mathbb{P}_c})$ is minimized during training. We optimize the codebooks with the loss

$$L_{HRQ}(x_s^{\mathbb{P}_c}) = \sum_{i=1}^{k} \Big( \|sg[r^i] - e^i\|^2 + \alpha\|r^i - sg[e^i]\|^2 \Big),$$

where $sg[\cdot]$ denotes the stop-gradient operator and $\alpha$ controls whether residuals are pulled toward the codebook vectors or vice versa. This objective ensures stable training and prevents codebook collapse.To allow backpropagation through the quantization step, the derivative with respect to $x_s^{\mathbb{P}_c}$ is modeled using the straight-through estimator, $\frac{dy_s^{\mathbb{P}_c}}{dx_s^{\mathbb{P}_c}} \approx I$. To solve conflicts between items in the representations generated by RQ, it adds an additional token that extends multitoken to uniquely identify the item. In practice, this is rarely necessary to uniquely identify an item as the multitokens from RQ most often suffice for the identification.

By embedding the residual quantization mechanism in hyperbolic space, HRQ directly exploits the curvature of the geometry to encode latent hierarchies. The resulting multitokens are more structured, semantically meaningful, and efficient than those produced in Euclidean space.

## 3.2 HYPERBOLIC RESIDUAL QUANTIZATION VAE (HRQ-VAE)

We introduce **Hyperbolic Residual Quantization VAE (HRQ-VAE)**, a generative model that integrates Hyperbolic Residual Quantization into an autoencoder framework. The method is inspired by RQ-VAE, which applies residual quantization in Euclidean space, but adapted to hyperbolic space, ensuring that the learned multitoken representations better align with latent hierarchies.

Formally, let $E_\theta : \mathbb{R}^d \to \mathbb{P}_c^{h_s}$ and $D_\theta^{\mathbb{P}_c} : \mathbb{P}_c^{h_s} \to \mathbb{P}_c^h$ denote the encoder and decoder networks, parameterized as hyperbolic neural networks (Ganea et al., 2018a). Since the input $x \in \mathbb{R}^d$ lies in Euclidean space, we map it to the hyperbolic manifold via the exponential map:

$$z^{\mathbb{P}_c} = E_\theta\big(\exp_0^c(x)\big).$$

HRQ quantizes the latent embedding: $([t^0, \ldots, t^{k-1}], y_s^{\mathbb{P}_c}) = HRQ_C(z^{\mathbb{P}_c})$,. The decoder reconstructs the embedding, which is mapped back to Euclidean space with the logarithmic map:

$$\hat{x} = \log_0^c \left( D_\theta^{\mathbb{P}_c}(y_s^{\mathbb{P}_c}) \right).$$

The model is trained with two objectives. The reconstruction loss $L_R(x) = \|x - \hat{x}\|^2$ encourages faithful reconstruction, while the quantization loss $L_{HRQ}(x_s^{\mathbb{P}_c})$ updates the codebooks and controls the interaction between residuals and code vectors. The total loss is $L(x) = L_R(x) + L_{HRQ}(x_s^{\mathbb{P}_c})$.

Optimization is performed with Riemannian SGD (Bécigneul and Ganea, 2018), ensuring that both codebook vectors and network parameters remain consistent within hyperbolic space. We denote the full process as

$$HRQ\text{-}VAE(x) = \left( [t^0, \ldots, t^{k-1}], \hat{x} \right).$$

By embedding HRQ within a hyperbolic autoencoder, HRQ-VAE learns multitoken representations that are discrete and natively hierarchical due to hyperbolic structure. The visualization of HRQ-VAE is shown in the Figure 3. The pseudocode for HRQ-VAE is in the Algorithm 1.

| $|C_i|$ | $k$ | Token type | Hidden dimensions | | | |
|---|---|---|---|---|---|---|
| | | | 4 | 8 | 16 | 32 |
| 64 | 3 | RQ | 71.2% | 73.7% | 69.8% | 67.1% |
| | | HRQ | **79.0%**(+10.9%) | **79.6%**(+8.0%) | **79.1%**(+13.3%) | **78.3%**(+16.7%) |
| | 4 | RQ | 71.2% | 70.9% | 70.3% | 64.6% |
| | | HRQ | **78.8%**(+10.7%) | **79.2%**(+11.8%) | **79.2%**(+12.6%) | **78.9%**(+22.1%) |
| 128 | 3 | RQ | 71.3% | 72.5% | 72.4% | 66.3% |
| | | HRQ | **79.5%**(+11.4%) | **79.5%**(+9.7%) | **79.5%**(+9.8%) | **79.1%**(+19.3%) |
| | 4 | RQ | 72.2% | 72.7% | 70.9% | 64.6% |
| | | HRQ | **79.1%**(+9.6%) | **79.4%**(+9.2%) | **79.6%**(+12.3%) | **78.7%**(+21.8%) |
| 256 | 3 | RQ | 72.4% | 73.2% | 71.2% | 66.2% |
| | | HRQ | **78.9%**(+9.0%) | **80.0%**(+9.3%) | **80.3%**(+12.9%) | **79.9%**(+20.7%) |
| | 4 | RQ | 73.5% | 72.2% | 70.9% | 67.3% |
| | | HRQ | **79.3%**(+7.9%) | **79.5%**(+10.1%) | **80.0%**(+12.9%) | **79.2%**(+17.7%) |

Table 1: Top 10 Recall of hypernymy prediction models trained on multitokens generated by RQ and HRQ. Despite operating on the same model and differing only in the structure of the multitokens, model that operated on HRQ multitokens produced significantly higher recall than model operating on RQ multitokens. The value $(+x.x\%)$ represents a percentage gain of HRQ w.r.t. RQ: (HRQ-RQ)/RQ. These results demonstrate that HRQ multitokens capture significantly more semantic information.

## 4 EXPERIMENTS

In this section, we empirically evaluate the quality of multitokens produced by RQ and HRQ. We evaluate the quality of tokens in a two-step pipeline. First, we learn the tokens for all entities. Then, we fix the tokens and investigate how well they perform in a downstream task.

We focus on data with latent hierarchies and our claim is that hyperbolic residual quantization produces better multitokens for data that contain latent hierarchies. Therefore, all our experiments are characterized by the clear existence of hierarchies in datasets. We evaluate HRQ in two distinct scenarios. First, we look at Hierarchy Modeling(Section 4.1), in which the multitokens are explicitly trained on the hierarchy and the downstream task is related to the hierarchy multitokens are modeling. The second scenario, which we call Hierarchy Discovery(Section 4.2), operates a dataset which contains latent hierarchies, but the model that learns multitokens is not supervised on these hierarchies. Furthermore, the downstream task is not directly related to the latent hierarchy as well.

Additionally, in Appendix D we inspect the structure of the space hypothesize on what causes benefits of HRQ multitokens. The implementation details for all methods are in Appendix C.

## 4.1 HIERARCHY MODELING

In this section, we investigate how effectively HRQ tokens capture hierarchical relationships compared to RQ. To do that, (H)RQ creates tokens by learning directly to predict hierarchical relation. Our experimental setup is similar to the main experiments from Nickel and Kiela (2017) that is adapted to discrete setting to compare the quality of discrete multitokens. Specficially, we use the transitive closure of the WordNet (Miller, 1995) noun taxonomy. The WordNet taoxnomy contains 82,115 nouns and 743,241 hypernymy relations. A hypernymy is a semantic "is-a" link where a general word(the hypernym) covers a group of more specific words (its hyponyms). We first learn the multitokens by simultaneously training an embedding and learning (H)RQ.

After we learn and create multitokens for all nouns, we fix multitokens, and we train a sequence-to-sequence transfomer model that translates noun to its hypernym, both represented with their corresponding multitokens. We evaluate the model by measuring recall@10 in the test dataset, which was not visible neither for the multitoken creation nor for the training of the sequence-to-sequence model. The test dataset is a randomly selected $15\%$ of all hypernymy relations. We learn the multitoken of the noun by embedding it in a continuous space, then contrastively pushing away nouns that are not in the hypernymy relation and pulling closer nouns that are. At the same time, the embedding is being quantized into multitokens by RQ or HRQ. Both embedding and codebook vectors are trained joinlty at the same time.

Formally, let $N$ be the set of nouns, and $H = \{(u,v) : u \in N, v \in N : u$ is a hypernym of $v\}$ be the set defining the hypernymy relation. Let $E_\theta$ be the $h$-dimensional embedding network, that embeds either in Euclidean or hyperbolic space depending on the model. Let $H'(u) = \{v : (u,v) \notin H\} \cup \{u\}$. We also have a (H)RQ algorithm with a codebook of length $k$, each codebook having $s$ vectors. Then the total loss for $(u,v) \in H$ is given by:

$$L(u,v) = \log \frac{e^{-d(E_\theta(u), E_\theta(v))}}{\sum_{v' \in H'(u)} e^{-d(E_\theta(u), E_\theta(v'))}} + L_{RQ}(E_\theta(u)) + L_{RQ}(E_\theta(v))$$

In practice, we limit $H'(u)$ to 50 sample nouns from $N$ that are not hypernyms of $u$ and $v$.

$L(u,v)$ is minimized for $\theta, C$ with $d$ being either the euclidean distance for RQ or hyperbolic distance for HRQ. As a result, it produces multitokens for all nouns $T(u) = [t_0, ..., t_{k-1}]$. In the next step, a transformer sequence-to-sequence model is trained to predict a hypernym for a given noun, both represented as their learned multitokens. The idea is that multitokens that better capture the structure of the space will serve as a more useful representation for the hypernymy generation.

To evaluate the representation quality of multitokens generated by RQ and HRQ, we investigate different combinations of parameters. We investigate the results for token lengths $k \in \{3, 4\}$. We vary the size of codebooks $s \in \{64, 128, 256\}$ and the dimensions of dense embeddings $h \in \{4, 8, 16, 32\}$. We focus on small dimensionalities of the dense embedding because usually before the residual quantization occurs, the embedding is mapped to a low-dimensional space.

The results for $k = 4$ and $|C_i| = 256$ are shown in the Figure 1. The complete results are shown in Table 1. Although the final sequence-to-sequence models differ only in the representations of the nouns and otherwise have the same architecture, the tokens generated by HRQ sometimes lead to an improvement of up to $20\%$ over the tokens generated by RQ. It clearly demonstrates the quality difference in favor of HRQ. The significant improvement is consistent across all dimensions tested. This demonstrates that HRQ is able to create significantly more semantic multitokens than the RQ, when it is trained to predict hierarchical relations.

## 4.2 HIERARCHY DISCOVERY

In real-world applications, the data often contains inherent hierarchical structures that are not explicitly labeled or available during model training. Although approaches directly supervised on the hierarchy can effectively learn to mimic known hierarchies, discovering latent hierarchical relationships without direct supervision presents a more challenging task. We call it the "Hierarchy Discovery", where the (H)RQ model creates hierarchical structures based solely on patterns present in the embeddings.

In this setting, we evaluate whether the hierarchical inductive bias of HRQ-VAE leads to multitokens that capture more semantic information compared to the standard RQ-VAE, when neither model has

access to hierarchical supervision during training. Both approaches must rely entirely on patterns within the embeddings themselves, but in HRQ-VAE there is an additional inductive bias towards the formation of hierarchical structures. Our evaluation focuses on downstream task performance as the primary measure of representation quality, reflecting the practical perspective that better representations should yield improved results on real-world problems.

We use the Amazon Reviews 2014 (McAuley et al., 2015) dataset, which contains a product catalog with detailed descriptions. We will generate multitokens of the products and then use them in a recommender system. We first generate dense embeddings with the MPNet (Song et al., 2020) from product descriptions, which serve as input for both our RQ-VAE and HRQ-VAE models.

RQ-VAE and HRQ-VAE are trained to produce multitokens without explicit hierarchical supervision. The quality of these discrete representations is subsequently measured by using them to train a sequence-to-sequence recommender system, where performance differences directly reflect the semantic richness captured by each quantization approach. The recommender system is a transformer encoder-decoder that predicts the next bought item based on all the previous history. Following the protocol of Rajput et al. (2023) we limit the user histories to those that have at least five items and truncate the histories to 20 items.

In order to test the model beyond the Amazon Reviews 2014 dataset, we also include the evaluation on the MovieLens 10M (Harper and Konstan, 2015) dataset. Note that both product and movies can be structured in latent taxonomical hierarchies, making them suitable for our case. As the MovieLens dataset does not contain the movie description, we first generate the descriptions with the LLM Claude (Anthropic, 2024). The prompt used to generate the description is included in the Appendix B.

| Dataset | Metric | Random | RQ-VAE | HRQ-VAE |
|---------|--------|--------|--------|---------|
| AR Beauty | NDCG@5 | 1.66%±0.07 | 2.29%±0.03 | **2.41%**±0.04 (+5.2%) |
| | Recall@5 | 2.06%±0.09 | 3.68%±0.03 | **3.74%**±0.04 (+1.6%) |
| | NDCG@10 | 2.35%±0.09 | 2.83%±0.05 | **2.89%**±0.05(+2.1%) |
| | Recall@10 | 3.87%±0.17 | 4.83%±0.06 | **5.01%**±0.06(+3.7%) |
| AR TaG | NDCG@5 | 1.51%±0.07 | 1.91%±0.02 | **1.94%**±0.02(+1.6%) |
| | Recall@5 | 1.97%±0.09 | 2.82%±0.03 | **2.93%**±0.03(+3.9%) |
| | NDCG@10 | 1.94%±0.09 | 2.45%±0.03 | **2.47%**±0.03(+0.8%) |
| | Recall@10 | 2.76%±0.16 | 4.22%±0.08 | **4.53%**±0.09(+7.3%) |
| AR SaO | NDCG@5 | 0.95%±0.07 | **1.03%**±0.02 | **1.03%**±0.02 (+0.0%) |
| | Recall@5 | 1.34%±0.08 | 1.58%±0.02 | **1.62%**±0.02(+2.5%) |
| | NDCG@10 | 1.29%±0.09 | **1.50%**±0.03 | 1.48%±0.02(−1.4%) |
| | Recall@10 | 2.41%±0.14 | 2.78%±0.04 | **2.85%**±0.04(+4.0%) |
| MovieLens | NDCG@5 | 11.42%±0.32 | 11.45%±0.20 | **11.76%**±0.24(+2.7%) |
| | Recall@5 | 17.43%±0.54 | 17.62%±0.21 | **17.90%**± 0.25(+1.6%) |
| | NDCG@10 | 13.21%±0.58 | 13.89%±0.28 | **14.27%**± 0.40(+2.7%) |
| | Recall@10 | 23.52%±0.73 | 25.11%±0.37 | **25.49%**± 0.33(+1.5%) |

Table 2: Results of recommender systems for different multitokens (Random, RQ-VAE and HRQ-VAE) across four datasets. The table reports average metric over 8 runs. The observed standard deviation is written on the right of the results. For the HRQ-VAE, percentage improvement over RQ-VAE is in the parantheses.

Apart from the RQ-VAE and HRQ-VAE we also include a baseline that consists of randomly sampled tokens with additional token that distinguishes conflicts, similarly to (H)RQ. The main results are shown in Table 2. The multitokens generated by HRQ-VAE consistently outperform RQ-VAE and the random baseline. The reported results are on the test set with each model type selected with the highest performance on the validation set.

## 5 RELATED WORK

**Quantized representations.** Quantized discrete representations are an alternative to dense embeddings, which recently gained popularity. The aim of a quantized representation is to create coarse information representations that focus on qualitative properties (Gray, 1984). Van Den Oord et al. (2017) proposes VQ-VAE that learns the vector codebook simultaneously together with the embeddings. This was further enhanced by RQ-VAE (Lee et al., 2022; Zeghidour et al., 2021) that calculates the sequence of discrete tokens by iteratively quantizing the residuals. Discrete representations are beneficial to use as labels, as they avoid issues of high-dimensional continuous generation. VQ-GANs (Esser et al., 2021; Yu et al., 2021) utilize vector quantization for adversarial (Goodfellow et al., 2020; Schmidhuber, 1991) image generation. Zeghidour et al. (2021); Yang et al. (2023) uses VQ-VAE for audio generation. RQ-VAE has been introduced both in audio (Zeghidour et al., 2021) and image processing (Lee et al., 2022).

**Hyperbolic Neural Networks.** Neural networks operating in hyperbolic space have been demonstrated to perform well in tasks and modalities with hierarchical structures. Sala et al. (2018); Nickel and Kiela (2017); Ganea et al. (2018b) demonstrate benefits of hyperbolic embeddings for data with latent hierarchies. Ganea et al. (2018a) derives multi-layer fully connected hyperbolic neural network. The benefits of utilizing hyperbolic neural networks can be observed in multiple areas containing hierarchies. Ma et al. (2021); Yang et al. (2024) model the taxonomy of objects in hyperbolic space. (Atigh et al., 2022; Khrulkov et al., 2020) applies hyperbolic nets to computer vision. Hyperbolic neural networks have shown their benefits in reinforcement learning (Cetin et al., 2022) due to the hierarchical nature of the unrolling episodes. Chamberlain et al. (2019); Chen et al. (2022); Sun et al. (2021) applies hyperbolic networks to recommender systems with two-fold motivation: 1) The bipartite graph nature of the interactions between users and items, which has been shown to correspond to a complex network (Krioukov et al., 2010), and 2) Taxonomical nature of the items. Hyper-VQ (Goswami et al., 2024) proposes vector quantization in hyperbolic space. It presents the quantization problem as a hyperbolic multinomial regression and is orthogonal to our contributions for HRQ-VAE. Both can be combined together and we consider that a promising future work. HIHPQ (**?**) introduces hierarchical hyperbolic product quantization, emphasizing the product space of hierarchical subspaces for large-scale retrieval. This stands in contrast to our approach, which focuses on modeling the hierarchical relationships between tokens themselves.

**Recommender Systems.** Traditional recommender systems represent items as ID-based discrete tokens and learn embeddings that capture user–item interactions Rendle et al. (2012); He et al. (2017); Kang and McAuley (2018). With the integration of large language models into recommendation Geng et al. (2022b); Li et al. (2024), there has been a shift toward content-based or structure-aware tokenization to better align with generative architectures. A notable development in this direction is the use of vector quantization. RQ-VAE has been applied to recommender systems Rajput et al. (2023), where continuous collaborative-filtering embeddings are discretized into tokens, enabling the training of Transformer-based sequential recommenders in the style of next-token prediction.

Apart from that sequential recommendation has been extensively studied through a diverse set of neural architectures that model user behavior patterns over time. Early deep sequential models such as GRU4Rec Hidasi et al. (2015) introduced recurrent networks for session-based recommendation, demonstrating that gated recurrent units can effectively capture short-term temporal dynamics. Convolutional approaches like Caser Tang and Wang (2018) encode item sequences as temporal "images" using vertical and horizontal convolutions to learn both union-level and point-level sequence features. Graph-based models, exemplified by HGN Ma et al. (2019), leverage hierarchical gating networks to capture high-order item transitions and disentangle long- and short-term preferences. With the rise of Transformer architectures, attention-based models such as SASRec Kang and McAuley (2018) and BERT4Rec Sun et al. (2019) became dominant, using unidirectional and bidirectional self-attention respectively to model sequential dependencies with flexible receptive fields. Subsequent work explored richer attention mechanisms, such as FDSA Zhang et al. (2019), which introduces feature-level attention to jointly model item and attribute interactions. More recent paradigms include self-supervised models like $S^3$-Rec Zhou et al. (2020b), which employ mutual information maximization to augment sequence representations with auxiliary self-supervised signals, and generative frameworks such as P5 Geng et al. (2022a), which cast recommendation tasks into a unified text-to-text format using pretrained sequence-to-sequence Transformers. Together,

these models illustrate the progression from recurrent and convolutional architectures to attention-based, self-supervised, and generative formulations that increasingly emphasize expressiveness, transferability, and compatibility with large-scale pretrained language models.

## 6 CONCLUSIONS AND FUTURE WORK

The results shown in this work indicate that HRQ-VAE creates hierarchical representations more robust than RQ-VAE, when latent hierarchies appear in the dataset. We show that even if the model is not directly supervised on the latent hierarchy, the multitoken generated by HRQ-VAE might still be more robust than multitoken generated by RQ-VAE. Due to ubiquity of latent hierarchies in practical dataset, we see potential for number of applications of HRQ-VAE.

Furthermore, improving the performance of discrete hierarchical tokens leads to more interpretable models, as the hierarchical tokens can be related to the data taxonomies. This direction of research might lead to models whose discrete representations remain robust under domain shifts and noisy inputs, leading to societal benefits such as enhanced transparency in AI-driven decision-making, and greater public trust through auditability of the deployed systems.

In this work, we limited the scope of investigation to datasets that exhibit clear latent hierarchies. However, RQ-VAE has shown impressive results in several domains that do not follow this assumption, such as image and audio processing. HRQ-VAE, after appropriate adaptation, can potentially be applied to these domains as well. Each modality presents its own unique challenges related to the scale of experiments, hyperbolic adaptations, and the analysis of performance-contributing factors. Due to these complexities, we considered these additional modalities outside the scope of the current paper. However, exploring the application of HRQ-VAE to these diverse domains remains an exciting direction for future work.

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

# A  HRQ-VAE

---

**Algorithm 1** HRQ-VAE

---

$\textbf{Input: } x \in \mathbb{R}^d$
$x^{\mathbb{P}_c} \leftarrow \exp_0^c(x)$
$x_s^{\mathbb{P}_c} \leftarrow E_\theta^{\mathbb{P}_c}(x^{\mathbb{P}_c})$
$y_s^{\mathbb{P}_c} \leftarrow 0, r_C^0 = x_s^{\mathbb{P}_c}$
$\textbf{for } i \in \{0, .., k-1\} \textbf{ do}$
$\quad e_C^i, t_i = q_C(r_C^i)$
$\quad$ add $t_i$ to the return sequence
$\quad r_C^{i+1} \leftarrow r_C^i \ominus_c e_c^{i-1}$
$\quad y_s^{\mathbb{P}_c} \leftarrow y_s^{\mathbb{P}_c} \oplus_c e_C^i$
$\textbf{end for}$
$y^{\mathbb{P}_c} \leftarrow D_\theta^{\mathbb{P}_c}(y_s^{\mathbb{P}_c})$
$y \leftarrow \log_0^c(y^{\mathbb{P}_c})$
$l_{\text{rec}} = ||x - y||^2$
$l_{\text{cmt}} = \sum_{i=0}^{k-1}(||sg[r_C^i] - e_C^i||^2 + \alpha||r_C^i - sg[e_C^i]||^2)$
$l \leftarrow l_{\text{rec}} + l_{\text{cmt}}$
$\nabla\theta \leftarrow \frac{dl}{d\theta} ; \nabla C \leftarrow \frac{dl}{dC}$
$\textbf{return } t_0, t_1, ..., t_{k-1}, \nabla\theta, \nabla C$

---

Algorithm 2: HRQ-VAE

# B  DATASETS DETAILS

## B.1  HIERARCHY MODELING

WordNet is a large, manually curated lexical database of English that groups words into synonym sets (synsets) and interlinks these synsets via semantic relations such as hypernymy and hyponymy, enabling rich hierarchical modeling of concepts (Miller, 1995). Each synset contains a gloss (brief definition) and example usages, and synsets are organized into noun, verb, adjective, and adverb hierarchies (**?**). For our hierarchy modeling, we focus exclusively on the noun subnetwork, where the "is-a" (hypernym) relation defines a directed acyclic graph representing a noun hierarchy.

The noun subnetwork consists of $82, 115$ nouns and $743, 241$ hypernymy relations. We split it into the train set and test set by randomly choosing $85\%$ of the hypernymy relations to be selected for the the train set. The Embedding, RQ and the sequence-to-sequence models are all trained on the train set. We use the remaining $15\%$ as the test set on which we report the performance.

## B.2  HIERARCHY DISCOVERY

We used four datasets to evaluate the HRQ-VAE performance in the Hierarchy Discovery section. Three data sets are the categories 'Beauty', 'Sports and Outdoors' and 'Toys and Games' from the Amazon Reviews 2014 suite (McAuley et al., 2015). We also evaluate HRQ-VAE on the MovieLens10M dataset (Harper and Konstan, 2015).

The (H)RQ-VAE uses dense embeddings of the items to learn the corresponding hierarchical tokens. In order to create dense embeddings of the items, we use a pretrained, fixed language model embedding (Song et al., 2020), which embeds the description of the item. The descriptions of the items are included in the Amazon Reviews 2014 datasets. For MovieLens, we first create the description from the movie title with the help of a Claude 3.5 Sonnet (Anthropic, 2024) language model.

In all experiments, we focus on predicting the next item the user interacted with (whether watched a movie or bought a product) and disregard the scores. This is a standard practice in the area of recommender systems (Rajput et al., 2023; Kang and McAuley, 2018; Zhou et al., 2020a).

In order to use MovieLens, we first create the descriptions with Claude 3.5 Sonnet (Anthropic, 2024). We use the following prompt to generate the movie description:

```
You are an expert in movie descriptions.  Your task is
to generate movie description that:
- contains a maximum of 100 words
```

| Dataset | Users | Items |
|---|---|---|
| AR Beauty | 22,363 | 12,101 |
| AR Toys and Games | 35,598 | 18,357 |
| AR Sports and Outdoors | 19,412 | 11,924 |
| MovieLens10M | 71,567 | 10,681 |

Table 3: Quantitative statistics of datasets used in Hierarchy Discovery experiments.

```
– captures the general theme of the movie and
 interesting specifics of the story
– can be used adequately in a search engine to
 search for a movie Your task is to generate a movie
 description for the following movie title.  Return the
 movie description and do not return anything else.
```

From the description, we generate a dense embedding in the same way as for AR datasets. In all datasets, we cut the histories shorter than 5 elements and limit the length of user histories to 20.

**Test/train split.** Following the standard evaluation (Rajput et al., 2023) method, we divide user histories into the test, validation, and training part with a leave-one-out strategy. If the user history is a sequence of items $[i_1, ..., i_T]$, with $T$ elements. The training set consists of history limited to $T - 2$ tokens. The validation set is a prediction of $i_{T-1}$ based on $[i_1, ..., i_{T-2}]$ and the test set is a prediction of $i_T$ based on $[i_1, ..., i_{T-1}]$. The last and second-to-last items are taken from all users for the validation and test split, regardless of the length trajectory. Note that $i_T$ in the notation above represents an item, not a token. Hence, for a multitoken scenario of tokens trained with (H)RQ-VAE, a single item $i_T$ will be represented by a multitoken of length $k$ and all $k$ atomic tokens will be selected for the test/validation set.

## C IMPLEMENTATION DETAILS

### C.1 HIERARCHY MODELING

**(H)RQ.** To create multitokens of nouns we learn at the same time the embedding of the nouns and the codebook that quantizes the tokens.

We investigate the results for token lengths $k \in \{3, 4\}$. We vary the size of the codebooks $s \in \{64, 128, 256\}$ and the dimensions of dense embeddings $h \in \{4, 8, 16, 32\}$. Other parameters follow Nickel and Kiela (2017). We use Stochastic Gradient Descent (Rumelhart et al., 1986) or Riemannian Stochastic Gradient Descent (Bonnabel, 2013) for the optimization of encoders and RQ/HRQ codebook respectively. We use the learning rate 1.0. We train both models for 1500 epochs, out of which first 20 epochs are warm-up epochs with learning rate equal to 0.01.

**Downstream Model.** The sequence-to-sequence model is trained to generate hypernyms of a noun, both represented as multitokens. Hence, both the input and the output of the model are a list of $k$ tokens from 0 to $s$. The transformer model has 4 layers for both the encoder and the decoder. The hidden

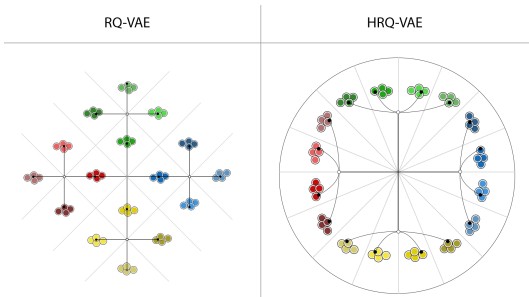

Figure 4: The embedding space structure induced by RQ-VAE and HRQ-VAE, respectively for a hierarchical tokens of length 2. The data is represented by coloured dots. Hue of the dot represents first hierarchical token. The shade represents second token. For the RQ-VAE the result is a typical effect of hierarchical clustering. HRQ-VAE due to exponential growth of the space has inductive bias to putting leaf nodes away on a similar distance away from the center.

dimension is equal to 256 with the feedforward dimension equal to 1024 and 8 attention heads. The embeddings of the encoder and decoder are tied. It is trained for 100 epochs with Adam (Kingma and Ba, 2014) optimizer with a learning rate equal to 0.001.

## C.2 Hierarchy Discovery

**(H)RQ-VAE.** The initial dense embedding of the text is calculated with 768 dimensional MPNET (Song et al., 2020). From the dense embedding, we train the (H)RQ-VAE and assign the new hierarchical token produced by the model to each item. The encoder in (H)RQ-VAE has 3 intermediate layers of size 512, 256, 128 with (H)ReLU activation and the output layer of size 32. The decoder has symmetric architecture to the encoder. The codebook has length 256 for each token and is not shared across tokens. We use batch size of 128, and train the (H)RQ-VAE for 5000 epochs with learning rates $[10^{-3}, 10^{-4}, 10^{-5}]$. We choose the learning rate that performed the best on the validation split of the downstream task and report the corresponding test result.

**Downstream Model.** We train the recommender system to evaluate the quality of the discrete representations produced by (H)RQ-VAE. User history is a sequence of items the user interacted with: either a movie they watched, or an item they bought. At each step, we predict the next item the user will interact with; specifically, we generate $k \in \{5, 10\}$ ranked guesses. To evaluate the quality of the set of guesses, we use two most popular recommender system metrics: Recall@K and NDCG@K. In our case of multitokens, each item is represented by a multitoken. Each user history has concanted multitokens of all items given a user bought(or a movie watched), and each specific recommendation is considered good if the entire multitoken corresponds to the true item the user interacted with. We split all the datasets into train,validation, and test set in the same way. We limit the histories to 20 interactions and filter the histories with less than 5 interactions. Furthermore, we select the last interaction as a test set, the second to last as a validation set, and everything else as a training set. We train a sequence-to-sequence transformer model (Vaswani, 2017) with T5 (Raffel et al., 2020) architecture. For each datapoint, an output sequence is the hierarchical representation of the next item, whereas the input is all their previous history. The model has a token embedding size of 384, 6 attention heads with 64 dimension each. and 1024 dimension of the feedforward net.

Our setup for hierarchy discovery follows the parameters of Rajput et al. (2023). However, the results differ significantly on the AR dataset. The fact that they differ consistently across all tokens and also across random baselines suggests that the cause of the inconsistency must lie in the final recommender system. However, after a detailed inspection and testing of different libraries, we were unable to reproduce the original results. However, please note that, contrary to Rajput et al. (2023) our claim is not about creating the best recommender system, but about comparing HRQ to RQ, and if the shift in the performance is caused by the downstream model - it is not important for our claim, as we use recommender system only as a downstream task to evaluate the quality of HRQ multitokens in comparison to RQ multitokens. All experiments were ran on a device equiped in a single 16GB Nvidia-V100 card.

|  | **RQ-VAE** | **HRQ-VAE** |
|---|---|---|
| **Variable** | $\|x_s\|_2$ | $\|\log_0^c(x_s^{\mathbb{P}^c})\|_2$ |
| **EV** | 0.7213 | 0.3251 |
| **Std. dev** | 0.2696 | 0.0664 |
| **CV** | 0.3738 | 0.2042 |

Table 4: Analysis of the norms for RQ-VAE and HRQ-VAE. We compare the euclidean norm of the low dimensional vector $x_s$ to the euclidean norm of hyperbolic $x_s^{\mathbb{P}^c}$ after mapping to the tangent space with logarithmic map. For the comparison we use the Coefficient of Variation defined as $CV(X) = \frac{\sigma(X)}{\mu(X)}$. It is used to compare the variability of a random variables with different orders of magnitude. RQ-VAE has almost twice the CV of HRQ-VAE which supports our claim about the structure of their corresponding spanning trees.

## D Structure of The Space

Suppose we have a set of points $S$ and we want to find a point that minimizes average distance to all points from $S$. In Euclidean space this point will be the center of mass of points, a simple average of all points from $S$. However, in hyperbolic space, the point that minimizes the average hyperbolic distance (Eq. 1) will be a continuous analogue to the nearest common ancestor node of all the nodes.

This leads to a vastly different structures when these spaces are clustered hierarchically and, as a consequence, to a vastly different structures for spanning trees of the residual quantization. In the Euclidean space the points corresponding to the leafs will be splattered around the space with the qunatization tree cutting into the centers of respective subclusters. Meanwhile, in the hyperbolic case, the leafs will be mostly spread around with the cluster "centers" being closer to 0 than the cluster points. This behavior has been observed in the hyperbolic clustering (Chami et al., 2020a). We visualize the structural differences in Fig. 4.

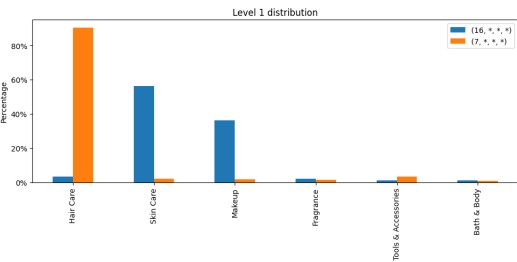

Figure 5: Example distribution of the first category in the ground-truth taxonomy conditioned on the first generated token (for first tokens 7 and 16). Token 7 has a concentrated distribution around Hair Care, whereas token 16 is split between Skin Care and Makeup. While this is different than the original hierarchy, the Skin Care and Makeup category are similar semantically, suggesting that HRQ-VAE discovered a different split than the original taxonomy.

This structure is beneficial for learning hierarchical relations for several reasons. Because the space is split radially most of the time the regions can have their own infinite part of the space, whereas in Euclidean division some regions are crammed close to center. As a consequence, the edges between regions are sharper than in the hyperbolic space, which might lead to poorer generalization. Finally, the structure imposed by the hierarchical euclidean quantization leads to strong utilization of the vector norms to select the cluster. On the other hand, hyperbolic quantization that leads to leafs being set the most outward in the spanning tree leaves the norm for the optimizer to choose, which can be an important benefit for gradient-based learning.

We argue that these structural difference of the hierarchical space of HRQ-VAE in comparison to space of RQ-VAE leads to the superior performance of HRQ-VAE in downstream tasks.

To quantitatively support this argument we inspect the norms of low-dimensional encoded representations $x_s$ and $x_s^{\mathbb{P}^c}$. Specifically, we argue that the norms will vary less in the hyperbolic space. To make a fair comparison we compare Euclidean norms, so the hyperbolic $x_s^{\mathbb{P}^c}$ vector is first transformed to the tangent space with logarithmic map. Moreover, as the models differ in the average norm we look at the Coefficient of Variation as a measure of interest. The coefficient of variation is defined for positive variables as $CV(X) = \frac{\sigma(X)}{\mu(X)}$. The results are shown in Table 4 and confirm that the norms vary significantly more for the vectors to be quantized in the euclidean space.

## E    MANUAL ANALYSIS

To better understand whether HRQ-VAE truly discovers hierarchical structure, we conduct several targeted manual inspections using the ground-truth taxonomy. The ground truth taxonomy is in the form of list of categories for each item. For example an item can have assigned categories ['Hair Care', 'Styling Products', 'Hair Sprays']. 'Styling Products' is a subcategory of 'Hair Care' and 'Hair Sprays' is a subcategory of 'Styling Products'. Importantly, this taxonomy is never used during training. We employ it only afterward to evaluate whether the learned tokens align with meaningful semantic organization.

We begin by examining specific cases of how a single learned first-level token partitions the data. For one such token, the induced distribution over ground-truth categories is usually very concentrated. When we compute the normalized entropy of these ground-truth labels weighted by the number of items assigned to this first generated token, we find it to be very low (around 0.17). For example, items that have first generated token equal to 7, almost all fall into Hair Care, with only small traces appearing elsewhere. This confirms quantitatively that the token corresponds to a clean, semantically coherent top-level branch, exactly the type of coarse split we expect HRQ to learn at the first stage. Another occuring case is when the items in a given token are split between two first categories. In our case, we observe that for token 16, where the elements are split between categories Skin Care and Makeup. These two dominant categories form a natural intersection in the ground-truth taxonomy, since many products (e.g., creams, skincare–makeup hybrids, cosmetic bundles) sit near their boundary. The learned token appears to capture precisely this shared region. In this case, the higher entropy reflects a meaningful intersection cluster, not noise.

The distribution of categories for items, which multitoken representations start with 7 and 16 and shown in Fig. 5.

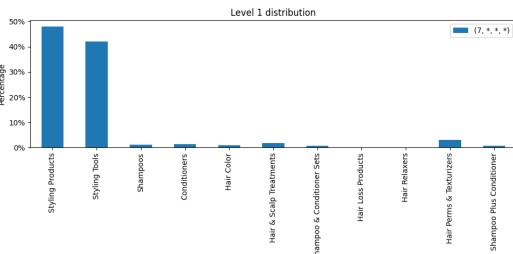

Figure 6: Second category of the true taxonomy conditioned for the items, which multitoken representation starts with 7.

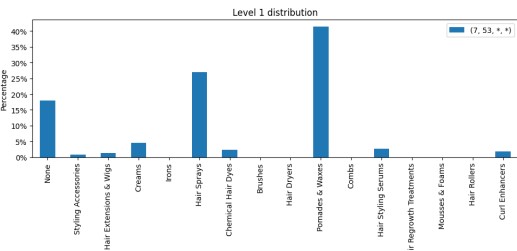

Figure 7: Third category of taxonomy for items, which multitokens start with [7, 53]. The choice of [7, 53] for demonstration was for illustrative purposes.

Next we investigate what happens to the second level category of the taxonomy, when conditioned on the first token. We investigate token 7 for the first category 'Hair Care' (the one that occurs most frequently for this token) and see (Fig. 6) that the second level category is characterized by higher indecisiveness, which is expected, as first token is supposed to give general split. However, we also observe, that the split is still more informative than the first category of original taxonomy. This also fits our expectations. This is because the original taxonomy used around 10 categories for the first level, whereas we have 256 tokens to use for the first level. Hence they might contain more information.

Having understood how the first token establishes coarse semantic regions, we next examine the behavior of the second-level token. We look at the distribution of third level of category (Fig. 7). At this depth, the distribution over leaf categories becomes more diffuse but still centers around a consistent family of styling-related leaves. A noticeable portion of items fall under a None label, reflecting missing ground-truth annotations. Manual inspection of these "None" cases reveals that they largely fit within the dominant semantic region, an example description of an item with "None" category that starts with [7, 53] is "[...] Get Beautiful hold while adding moisture and shine to natural hair. - Perfect For locks and twists. [...]", which fits the general theme or hair stylizers. This shows that HRQ-VAE is organizing the space semantically consistently and in cases like that could be even used for the enhancement of the original, noisy taxonomy. We consider this an interesting future work.

Altogether, these inspections show a clear progression: the first generated token creates low-entropy, high-level branches. The next token refines them into coherent sub-branches and deeper tokens capture fine-grained semantic distinctions that remain aligned with the underlying data. This provides strong qualitative evidence that HRQ-VAE is discovering meaningful hierarchical structure, not merely improving performance incidentally.

## F  LIMITATIONS

Although HRQ and HRQ-VAE demonstrate better performance in the discussed tasks, they come with some limitations. The biggest limitation is the strong assumptions about the type of data. Currently, we limit the claim to the situation where the dataset has latent hierarchies, and at the same time, we are interested in discrete representations. This is a very specific situation. Extending the evaluation to domains of general application in which RQ-VAE succeeded, such as image or audio, would

greatly increase the influence. However, the current version does not investigate performance in this direction.

A second limitation arises from the practical constraints of using hyperbolic neural networks in the HRQ-VAE component. Training models in hyperbolic space often requires higher numerical precision to maintain stability. Most notably, HRQ-VAE similarly to other hyperbolic models needs to run in float64 rather than float32, as otherwise the training becomes unstable. This increases computational cost, memory usage, and training time, and may limit scalability for very large models or large-batch training regimes.

## G  REPRODUCIBILITY STATEMENT

To ensure reproducibility, we report standard deviations in Table 2. In addition, we provide detailed descriptions of our models and implementation choices, including the exact prompt used to generate the movie title descriptions for the Movielens dataset, in the Appendix.

## H  LLM USAGE

In preparing this manuscript, large language models (LLMs) were employed solely as writing assistants to polish the text. Their role was limited to improving readability, grammar, and style, without contributing to the development of ideas, the design of experiments, the analysis of results, or the generation of original content. All scientific contributions, including conceptualization, methodology, and interpretation, are entirely the authors' own.

