# OpenReview forum: "Hyperbolic Residual Quantization: Discrete Representations for Data with Latent Hierarchies"
_ICLR.cc/2026/Conference — Submitted to ICLR 2026_

### Official Review · Reviewer_6yTa · 2025-10-31

**Soundness:** 2
**Presentation:** 2
**Contribution:** 2
**Rating:** 2
**Confidence:** 3

**Summary:**

This paper introduces Hyperbolic Residual Quantization (HRQ), a new method for creating discrete, multi-token representations of data that has a latent hierarchical structure. The authors argue that standard Residual Quantization (RQ) is suboptimal because its reliance on Euclidean geometry is a poor match for the branching, tree-like nature of hierarchies. The proposed HRQ method adapts this process to a hyperbolic manifold, performing residual calculations and distance measurements using hyperbolic operations. The method is evaluated in two settings: a "Hierarchy Modeling" task using WordNet hypernyms and a "Hierarchy Discovery" task where tokens are learned for a downstream recommender system. In both scenarios, the discrete tokens generated by HRQ are shown to be more effective and lead to better downstream performance than those generated by the standard Euclidean RQ.

**Strengths:**

The paper's core idea is a significant strength. The motivation to move from Euclidean to hyperbolic geometry for hierarchical data is well-founded, logical, and represents a novel application of these concepts to residual quantization.

The empirical results in the supervised "Hierarchy Modeling" task seems very strong. The reported performance gains over the baseline RQ method are substantial and provide clear evidence that the HRQ method is more effective at capturing the target hierarchical relationships.

The paper is also well-written and clear. It does a good job of introducing the necessary background concepts in both hyperbolic geometry and residual quantization, making the new method and its motivation accessible.

**Weaknesses:**

A primary weakness is an apparent contradiction in the method's technical implementation. While the paper's premise is to use native hyperbolic operations, the loss function for training the codebook (l_cmt in Algorithm 1) appears to be a standard Euclidean L2 norm. This is not justified and seems to conflict with the core motivation of avoiding Euclidean metrics. I have not gone through all math to check if this holds for hyperbolic space (thus lowered my confidence). Still this seems like a major contradiction.

A second major weakness is the omission of key, highly relevant baselines. The paper does not compare against other state-of-the-art models for hierarchical representation, such as the Hierarchy Transformer (HiT). This is a critical omission because HiT shares the same goal (encoding hierarchies), uses the same solution space (hyperbolic geometry), and is evaluated on the same key dataset (WordNet). The current paper only compares HRQ to Euclidean RQ, which merely proves that a hyperbolic approach is better than a Euclidean one for this specific task. It fails to demonstrate that this two-stage quantization-then-model approach is competitive with, or superior to, end-to-end continuous hyperbolic models.

The claims made in the "Hierarchy Discovery" experiment are also not fully substantiated. The experiment shows that HRQ tokens lead to better performance in a recommender system, which the authors attribute to the model's discovery of latent hierarchies. However, no direct evidence is provided that a meaningful hierarchy was actually found. The improved performance could just as easily be a result of the hyperbolic autoencoder acting as a more effective regularizer for that specific dataset.

**Questions:**

Could you clarify the choice of using a Euclidean L2 norm for the codebook loss (l_cmt) instead of a loss function based on the hyperbolic distance metric? Did you experiment with a fully hyperbolic loss, and if so, how did it affect training stability and final performance?

For the "Hierarchy Discovery" experiment, do you have any way to qualitatively or quantitatively verify that the model is actually learning a semantically meaningful hierarchy, beyond the improved downstream task performance?

How do you think your two-stage quantization-then-translation approach would compare to end-to-end models like the Hierarchy Transformer (HiT), which also use hyperbolic geometry to learn continuous representations directly on the WordNet task?

What is your hypothesis for why the standard RQ baseline performs so poorly as the hidden dimensionality increases in Figure 1, while HRQ's performance is so stable?

Could you provide more geometric intuition on the hyperbolic residual calculation? In Euclidean space, the residual is a new vector. What is the geometric interpretation of this operation in the Poincaré ball model?

---

> ### Author Response · Authors · 2025-11-28
> **Rebuttal Answer 1/3**
>
> We sincerely thank the reviewer for their careful reading of our paper and for the very thoughtful and technically informed comments. While the overall scores are modest, we truly appreciate that the review engages deeply with both the geometric motivation and the practical implementation details of HRQ, and raises precisely the kinds of questions that demonstrate in-depth reading of the manuscripts. Below, we address each of these points in turn.
>
> ### Q1:
>
> > While the paper's premise is to use native hyperbolic operations, the loss function for training the codebook (l_cmt in Algorithm 1) appears to be a standard Euclidean L2 norm. [….]  Could you clarify the choice of using a Euclidean L2 norm for the codebook loss (l_cmt) instead of a loss function based on the hyperbolic distance metric? Did you experiment with a fully hyperbolic loss, and if so, how did it affect training stability and final performance?
>
> Thank you for this insightful question. It touches precisely on the dilemma we faced during early experimentation. We did in fact test both variants in the preliminary results and we observed that euclidean loss leads to more stable results. Importantly, because the distance is calculated between the closest matching vectors, the conceptual discrepancy is smaller than it may appear: the largest differences between Euclidean and hyperbolic distances arise for large distances, where curvature strongly stretches space. In HRQ, however, the loss is computed only over the top-k nearest codewords. These points lie in a small local neighborhood of the basepoint, where hyperbolic and Euclidean metrics behave very similarly. Only in occasional edge cases, when a selected codeword happens to be far from the target, does curvature meaningfully affect the distance. Our initial results suggested that aggressively optimizing these rare edge cases harmed global training stability more than it helped, so we opted for the simpler loss.
>
> Finally, we note that mixing Euclidean distances with hyperbolic representations is not unprecedented. Similar hybrid choices have been shown to be empirically advantageous. For example, in Hyperbolic Neural Networks [1], Table 1 reports that the Euclidean MLR head atop a hyperbolic GRU+FFN outperforms the fully hyperbolic alternative (We also refer to the great tutorial on hyperbolic neural networks, which explains it very well: https://youtu.be/JHN_AKi25wU at around 2:30:40). Our choice follows the same practical rationale: a fully hyperbolic geometry is conceptually elegant, but hybrid Euclidean losses can deliver more robust training while preserving the benefits of hyperbolic representation in this particular case.
>
> ### Q2:
>
> > For the "Hierarchy Discovery" experiment, do you have any way to qualitatively or quantitatively verify that the model is actually learning a semantically meaningful hierarchy, beyond the improved downstream task performance?
>
> Thank you for raising this point. Several reviewers requested a more explicit demonstration of the hierarchies learned in the discovery stage. In response, we have added new visualizations in the section E of the appendix to the revised paper that directly compare the HRQ-VAE latent hierarchy to the *ground-truth taxonomy* of the Amazon Reviews 2014 dataset. Please note, that this taxonomy was **never used during training** and serves solely as an external reference for qualitative and quantitative validation.
>
> The new plots reveal a clear alignment between the discovered structure and the true product categories: items belonging to the same or closely related categories cluster in nearby branches, and higher-level groupings emerge naturally from the discrete HRQ tokens. This alignment provides direct evidence that HRQ-VAE is not only improving the downstream recommender task, but is genuinely recovering meaningful semantic hierarchies in an unsupervised manner.

---

> > ### Author Response · Authors · 2025-11-28
> > **Rebuttal Answer 2/3**
> >
> > ### Q3:
> >
> > > How do you think your two-stage quantization-then-translation approach would compare to end-to-end models like the Hierarchy Transformer (HiT), which also use hyperbolic geometry to learn continuous representations directly on the WordNet task?
> >
> > Thank you for this question, it directly touches on the motivation for introducing *quantization* in the first place, namely the need to **generate** the learned representations as tokens.
> >
> > Models like HiT operate in a fully continuous regime: they encode nodes into a continuous hyperbolic space and are then trained to score or classify hierarchical relations (e.g., parent–child links in WordNet). This is a very natural and powerful setup for *discriminative* hierarchy modeling. By contrast, our primary motivation is different: in the **hierarchy discovery** setting, the learned representations must also serve as **labels for an autoregressive generator** (e.g., a sequence model that predicts discrete tokens). If one tries to use continuous vectors as labels and learns them with an L2-type regression objective, the generator tends to predict conditional *means* of multimodal distributions. In a bimodal case, for instance, the predicted vector collapses to the average between two valid modes, which corresponds to no meaningful discrete state. Avoiding this “regression-to-the-mean” issue with continuous labels typically requires more complex generative machinery (GANs, diffusion, flows), which is difficult to integrate with large-scale sequential models. The standard and scalable workaround in practice is to **quantize** and let the generator predict *discrete* codes.
> >
> > Our work is precisely about making this quantization step *hierarchy-aware*: HRQ provides discrete, multi-token representations that respect hyperbolic geometry and can be directly generated by an autoregressive model. While the WordNet task *could* be phrased as a classification problem over all candidate parents, we deliberately formulate it as a **generation** problem to match the intended use case of our method. In real hierarchy-discovery settings, the model must output the representation itself, rather than score a predefined set of candidates.
> >
> > Regarding a direct comparison: on a purely discriminative WordNet task, if HiT is phrased as a classification model over (noun, candidate parent) pairs and allowed to score all candidates, a strong continuous model like HiT would probably even outperform HRQ, since quantization is inherently lossy. However, this comes at a computational cost (requiring forward passes over all candidate parents) and does not address the generative use case where the representation itself must be *produced* token-by-token. In that generative regime, a hierarchy-aware quantization scheme like HRQ has a clear advantage: it yields discrete, semantically structured labels that can be predicted directly, whereas continuous HiT-style representations trained with L2 objectives would be prone to the aforementioned collapse-to-the-mean behavior when used as generation targets.
> >
> > ### Q4
> >
> > > What is your hypothesis for why the standard RQ baseline performs so poorly as the hidden dimensionality increases in Figure 1, while HRQ's performance is so stable?
> >
> > Once again, this is an excellent question, and it connects directly to the reasoning in the previous answer: in our setting, the learned representations are used not only as embeddings but also as **labels** for generation (here we also note that while we learn the representations, we make sure they are unique (lines 251-253 from the manuscript), so the model can’t “cheat” by learning non-unique representations). Because of this, the usual expectation that larger hidden dimensions should lead to better performance no longer reliably applies.
> >
> > Our hypothesis is that in Euclidean RQ, increasing dimensionality gives the codebook too much freedom, making the representations less structured and therefore harder for the autoregressive model to predict. In contrast, HRQ benefits from the **intrinsic hierarchical bias of hyperbolic space**, which encourages a stable tree-like organization regardless of the dimensionality. This built-in geometric structure makes HRQ far more robust to changes in dimension, whereas Euclidean RQ becomes less predictable as the space grows.

---

> > > ### Author Response · Authors · 2025-11-28
> > > **Rebuttal Answer 3/3**
> > >
> > > ### Q5
> > >
> > > > Could you provide more geometric intuition on the hyperbolic residual calculation? In Euclidean space, the residual is a new vector. What is the geometric interpretation of this operation in the Poincaré ball model?
> > >
> > > In Euclidean space, the residual tells you how to move from the codeword c to the point x: you take a straight step on a flat surface. In the Poincaré ball, the idea is exactly the same, except the “surface” is curved. The shortest way to get from c to x is no longer a straight line but a curved path. The hyperbolic residual simply describes **that** movement: the direction you need to go on the manifold and how far along that curved path you must travel to reach x.
> > >
> > > Each residual in residual quantization can be thought of as refining or specifying the concept represented by the previous one. In hyperbolic space. which can be viewed as a continuous analogue of a tree. this refinement naturally corresponds to **branching out** from a more general node toward a more specific one. The residual therefore captures *which branch to follow next* and *how much specialization* is needed.
> > >
> > > Finally, unlike the commitment loss case (where Euclidean and hyperbolic versions behave similarly for small distances), the residual calculation genuinely needs to be carried out in hyperbolic geometry. If we were to compute residuals using Euclidean subtraction inside the Poincaré ball, the resulting vectors would collapse to extremely small magnitudes because Euclidean differences severely underestimate hyperbolic distances, especially near the boundary. Hyperbolic residuals avoid this collapse by correctly “stretching” movements to their true hyperbolic scale - to the size those displacements would have if anchored at the center of the space. This ensures that each residual meaningfully represents a step along the hierarchical structure encoded by the manifold.
> > >
> > > We thank the reviewer for the detailed feedback and the insightful technical questions. We have addressed each of these points carefully in our responses above and incorporated the corresponding clarifications and additions into the revised manuscript. We also appreciate the reviewer’s emphasis on the paper’s significant strengths, such as strong motivation for hyperbolic quantization, the clarity of writing, and the empirical performance, which we are glad came through clearly.

---

### Official Review · Reviewer_rJFw · 2025-11-01

**Soundness:** 2
**Presentation:** 2
**Contribution:** 2
**Rating:** 2
**Confidence:** 3

**Summary:**

HRQ replaces Euclidean RQ with hyperbolic geometry—embedding, residuals, and distances—all aligned to hierarchical branching. On WordNet (supervised hierarchy modeling) and hierarchy discovery tasks, its discrete multitoken representations consistently outperform Euclidean RQ, with gains up to ~20% on the former.

**Strengths:**

Across hierarchy modeling (WordNet) and hierarchy discovery tasks, HRQ’s tokens improve performance—up to ~20% in the supervised setting compared to RQ.

**Weaknesses:**

- Insufficient comparisons to strong hyperbolic baselines. Prior work on hyperbolic quantization (e.g., *HyperVQ*) is discussed but not empirically compared. As a result, the claimed advantages over stronger or more numerous hyperbolic baselines remain unsubstantiated; the study lacks both breadth in baselines and depth in experimental analysis.
- Unclear motivation for hyperbolic space. The paper does not convincingly quantify *why* hyperbolic geometry is necessary here. The authors’ claim — *“If multitokens are structured semantically, they can share tokens across compositions, enabling information sharing and more efficient, robust representations than flat tokens.”* — is not backed by sufficient **theoretical** or **empirical** evidence.
- Limited validation scope. The method is suggested to be evaluated on more datasets and more hierarchical tasks to establish generality and strengthen the paper’s empirical foundation.

**Questions:**

1. How do the resulting discrete token representations perform when incorporated into Transformers and downstream tasks, compared to Euclidean ones?
2. Are there any case studies that demonstrate the learned semantic hierarchy?

---

> ### Author Response · Authors · 2025-11-28
> **Rebuttal Answer 1/2**
>
> We appreciate the reviewer’s feedback. We are encouraged that the reviewer found clear strengths in our empirical results, particularly the consistent improvements of HRQ-VAE over Euclidean RQ across hierarchy modeling and hierarchy discovery tasks. Below, we clarify the motivation and methodological choices, address the questions raised, and provide additional analysis that strengthens the paper.
>
> > insufficient comparisons to strong hyperbolic baselines. Prior work on hyperbolic quantization (e.g., *HyperVQ*) is discussed but not empirically compared.
>
> There are two reasons we did not compare to HyperVQ
>
> 1. HyperVQ is not a competing method to HRQ-VAE. HRQ-VAE consists of calculating the residuals and vector quantization. For vector quantization we used “naive adaptation” of VQ to the hyperbolic space, but with HyperVQ codebase publicly available, we could use HyperVQ as a base for hyperbolic quantization and combine it with the residual calculation necessary for RQ. The combined method would probably improve the results even further instead of being a competitor.
> 2. HyperVQ is highly technical, it formulates VQ as a hyperbolic Multinomial Logistic Regression, yet it does not have public code implementation. Hence, reimplementing the method seems like a costly endeavour that while might improve our results, does not help with the validation of our claim.
>
> > Unclear motivation for hyperbolic space. The paper does not convincingly quantify *why* hyperbolic geometry is necessary here. The authors’ claim — *“If multitokens are structured semantically, they can share tokens across compositions, enabling information sharing and more efficient, robust representations than flat tokens.”* — is not backed by sufficient **theoretical** or **empirical** evidence.
>
> These are two separate points, one is the motivation for the hyperbolic space and the second is the claim about benefits of semantically structured multitokens, which is not related to the hyperbolic space. Let’s start with the hyperbolic space:
>
> The motivation for hyperbolic space is pretty straghtforward and based on the fact that we assume data to have latent hierarchies (represented as trees). Hyperbolic space induces hierarchical inductive bias with the curvature of the space. For example it grows exponentially with radius, which allows to embed a tree isometrically. Hence, hyperbolic space is more suitable when operating on tree-like structured data in the latent space. This type of reasoning has been a base motivation for multiple [1,2,3] (most) publications that combine hyperbolic space and neural network, and has been also demonstrated theoretically [4,5]. This discussion is written out in the introduction of the manuscript and furthermore, we also would like to point out that other reviewers found the motivation behind using hyperbolic space for our setting clear and well founded.
>
> > *“If multitokens are structured semantically, they can share tokens across compositions, enabling information sharing and more efficient, robust representations than flat tokens.”*
>
> Our statement regarding information sharing in multitoken representations follows directly from the construction: when a concept is represented by a sequence of codes, shared prefixes (or subsequences) allow related concepts to reuse part of their representation. This is a structural property of multitoken composition rather than a consequence of the geometric space used.
>
> The motivation for hyperbolic space is independent of that observation. We do not argue that hyperbolic geometry is *necessary* for multitoken models; rather, we explore whether the known advantages of hyperbolic embeddings for hierarchical or tree-like data can benefit multitoken decompositions that naturally introduce hierarchical relationships between residuals - and therefore tokens. Our experiments show that incorporating hyperbolic geometry yields measurable improvements, which is the empirical evidence supporting the choice.
>
> > Limited validation scope. The method is suggested to be evaluated on more datasets and more hierarchical tasks to establish generality and strengthen the paper’s empirical foundation.
>
> We appreciate the reviewer’s suggestion to extend the empirical evaluation further. At the same time, we note that our current experiments already cover a broader set of hierarchy-related tasks than what is standard in this line of work. Specifically, we evaluate on four established hierarchy-discovery datasets and additionally include hierarchy-modelling experiments, while prior work such as TIGER, which our method builds upon, was evaluated on only three hierarchy-discovery datasets that form a subset of our evaluation. Given this comparison to widely accepted baselines, we believe our current evaluation provides a solid and representative empirical foundation.

---

> > ### Author Response · Authors · 2025-11-28
> > **Rebuttal Answer 2/2**
> >
> > > How do the resulting discrete token representations perform when incorporated into Transformers and downstream tasks, compared to Euclidean ones?
> >
> > The downstream performance of the discrete token representations is directly evaluated in our hierarchy-discovery experiments. In this setting, we incorporate the HRQ-VAE multitoken representations into an autoregressive Transformer and compare them with representations obtained from a standard Euclidean RQ-VAE. The results show that HRQ-VAE tokens perform well and achieve consistent improvements, which confirms that the learned discrete codes transfer effectively to Transformer-based models.
> >
> > This downstream use case is in fact the primary application motivation for our method.
> >
> > > Are there any case studies that demonstrate the learned semantic hierarchy?
> >
> > We have now added a case study (section E of the appendix) that demonstrates the learned semantic hierarchy. In this new analysis, we visualize the hierarchical structure produced by HRQ-VAE and compare it to the *true* label taxonomy from the Amazon Reviews 2014 dataset. Importantly, these label hierarchies are **not** used during training; they are only employed for post-hoc evaluation. The comparison shows that the learned multitoken structures align well with the underlying semantic categories: shared prefixes correspond to higher-level classes, while deeper branches reflect more fine-grained distinctions.
> >
> > [1] - **Nickel & Kiela**  - *Poincaré Embeddings for Learning Hierarchical Representations*
> >
> > [2] - **Ganea et al.-** *Hyperbolic Neural Networks*
> >
> > [3] - **Shimizu et al. -** *Hyperbolic Neural Networks++*
> >
> > [4] - **Sarkar -**  *Low Distortion Embeddings of Trees in Hyperbolic Space*
> >
> > [5] - **Gromov** - Hyperbolic Groups

---

### Official Review · Reviewer_brqf · 2025-11-01

**Soundness:** 2
**Presentation:** 2
**Contribution:** 2
**Rating:** 4
**Confidence:** 3

**Summary:**

This paper proposes HRQ, a method for learning discrete multitoken representations for hierarchical data structures.

**Strengths:**

It demonstrates empirical improvements, with HRQ tokens outperforming standard residual quantization on multiple benchmark tasks related to hierarchy modeling and discovery

**Weaknesses:**

Regarding the references, the authors have omitted numerous relevant citations, including HiHPQ and works on knowledge-graph representation and recommender systems, among others.

For the main loss formula, the authors failed to provide proper numbering, which represents poor writing practice. The authors employ Euclidean distance for RQ in Euclidean space; however, Euclidean RQ typically utilizes dot product. Furthermore, the contrastive loss formula is not an essential component of RQ and should only be employed when specific semantic hierarchical supervision modeling is required.

HRQ primarily compares against the Euclidean version of RQ, lacking systematic analysis relative to other state-of-the-art structured representation learning approaches, and missing broader baseline comparisons.

The paper exhibits limited theoretical analysis and explanation, relying predominantly on experimental results while lacking theoretical discussion and proof regarding the limitations of hyperbolic space quantization mechanisms.

**Questions:**

The manuscript does not link to code or a reproducible pipeline. Is a public implementation planned, and can you outline what main dependencies and practical steps are required to reproduce your experiments?

---

> ### Author Response · Authors · 2025-11-28
> **Rebuttal answer 1/2**
>
> We thank the reviewer for their thoughtful feedback and for highlighting the practical value of our approach, in particular its ability to learn structured discrete representations that improve performance across several hierarchy-focused tasks.
>
> It seems to us that there might have been a confusion about the setup. Specifically, we would like to clarify that HRQ-VAE assumes that there is no given hierarchy, or any type of knowledge-base given. The goal of the task is to create a discrete representation from a dense representation that can later be used in a downstream task, most often as a label, because modelling continuous vectors directly is problematic (please see the answer to the question 3 to the reviewer 6yTa for more details). Below we address specific issues separately.
>
> ### W1:
>
> > HRQ primarily compares against the Euclidean version of RQ, lacking systematic analysis relative to other state-of-the-art structured representation learning approaches, and missing broader baseline comparisons.
>
> This question is one of the reason we suspect the confusion we listed. Our setup does not involve any structure, neither in the form of taxonomy or knowledge base, hence structured representation learning approaches are not applicable. We have a quite specific setting in which we learn the tree representation and both the hyperbolic space and RQ-VAE induce the taxonomical, hierarchical representation. For example, we evaluated HIHPQ suggested by the reviewer in the hierarchy modelling task, and it underperforms in comparison to RQ-VAE. Specifically:
>
> |  | 4 | 8 | 16 | 32 |
> | --- | --- | --- | --- | --- |
> | HRQ-VAE | 79.3% | 79.5% | 80.0% | 79.2% |
> | RQ-VAE | 73.5% | 72.2% | 70.9% | 67.3% |
> | HIHPQ | 66.5% | 67.1% | 66.8% | 66.2% |
>
> this is expected, as RQ-VAE induces hierarchical structure by operating on the residuals. HIHPQ operates on **product** space, and while it does induce hierarchy inside these subspaces by having hyperbolicity of the subspaces - the product nature of the combined space produces euclidean, non-hyperbolic relation between them.
>
> ### W2:
>
> > Regarding the references, the authors have omitted numerous relevant citations, including HiHPQ and works on knowledge-graph representation and recommender systems, among others.
>
> We extended the related work by the HiHPQ and substantially extended recommender system related work section. Regarding knowledge-graph representations - as we stated above, we believe this comment comes from a misunderstanding about our setup. We don’t believe knowledge-graph representation learning is relevant to our problem.

---

> ### Author Response · Authors · 2025-11-28
> **Rebuttal Answer 2/2**
>
> ### W3:
>
> > For the main loss formula, the authors failed to provide proper numbering, which represents poor writing practice.
>
> We kindly ask the reviewer to clarify what numbering in their opinion should be used. We followed the standard practice of numbering the equations that are used **later** in the text (for example we numbered the equation (1) that defines the hyperbolic distance, because it is referred to later in the text - in the appendix D. ). On the other hand, the equations that are referred to only directly under them, should not be numbered. We refer to the main loss only directly under it.
>
> > The authors employ Euclidean distance for RQ in Euclidean space; however, Euclidean RQ typically utilizes dot product.
>
> This is not true. We kindly refer the reviewer to the original RQ-VAE [1], equation 6 and 7, or TIGER [2]  (which applies RQ-VAE to the recommender system setup), top of page 5. Both of them utilize Euclidean distance, and not dot product.
>
> > Furthermore, the contrastive loss formula is not an essential component of RQ and should only be employed when specific semantic hierarchical supervision modeling is required.
>
> That’s why it is not used in the Hierarchy Discover experiments. It is used in the Hierarchy Modelling experiments, because this setup follows Nickel & Kiela [3] setup in a quantized version, and there it was modelled contrastively.
>
> > The paper exhibits limited theoretical analysis and explanation, relying predominantly on experimental results while lacking theoretical discussion and proof regarding the limitations of hyperbolic space quantization mechanisms.
>
> The theoretical motivation for the paper is similar to other papers applying hyperbolic space to dense embeddings that contain latent hierarchies [3, 4, 5]. The theoretical foundation behind hyperbolic space being favourable for tree representation over euclidean space is also well known [6, 7].
>
> > The manuscript does not link to code or a reproducible pipeline. Is a public implementation planned, and can you outline what main dependencies and practical steps are required to reproduce your experiments?
>
> At the moment the source code is a proprietary software. We used HypLL library https://hyperbolic-learning-library.readthedocs.io/en/latest/ for hyperbolic space that was combined with RQ-VAE implementation. https://github.com/lucidrains/vector-quantize-pytorch.
>
> [1] Lee, Doyup, et al. "Autoregressive image generation using residual quantization."
>
> [2] Rajput, Shashank, et al. "Recommender systems with generative retrieval."
>
> [3] Nickel, Maximillian, and Douwe Kiela. "Poincaré embeddings for learning hierarchical representations.”
>
> [4] Ganea, Octavian, Gary Bécigneul, and Thomas Hofmann. "Hyperbolic neural networks.”
>
> [5] Sala, Frederic, et al. "Representation tradeoffs for hyperbolic embeddings.”
>
> [6] Gromov, Mikhael. "Hyperbolic groups.”
>
> [7] Sarkar, Rik. "Low distortion delaunay embedding of trees in hyperbolic plane." *International symposium on graph drawing*

---

### Official Review · Reviewer_cMFE · 2025-11-03

**Soundness:** 3
**Presentation:** 4
**Contribution:** 3
**Rating:** 8
**Confidence:** 3

**Summary:**

The authors propose Hyperbolic Residual Quantization (HRQ), a novel discrete representation learning technique that performs residual quantization in hyperbolic space. They propose this method to better model hierarchical data structures, which are poorly captured by traditional Euclidean-based quantization methods. By leveraging the exponential growth and tree-like properties of hyperbolic geometry, HRQ produces multi-token representations that more accurately reflect latent hierarchies in data such as WordNet or product catalogs.

**Strengths:**

The paper introduces a novel integration of hyperbolic geometry into residual quantization, enabling discrete representations that naturally capture hierarchical relationships.

Theoretical motivation is strong and well-founded, aligning hyperbolic geometry’s exponential growth with hierarchical data’s inherent structure.

Experiments show consistent and significant improvements over Euclidean baselines in both supervised and unsupervised settings, demonstrating the model’s robustness and practical relevance.

**Weaknesses:**

The method assumes data follows a hierarchical structure, limiting its usefulness in domains without clear hierarchies.
How does the model behave when applied to data lacking an explicit or strong hierarchical organization?


The computational cost and stability issues of hyperbolic training are not explored, leaving efficiency and scalability uncertain.
What are the practical training challenges or trade-offs when scaling HRQ to large datasets or deeper architectures?

**Questions:**

Same as weakness

---

> ### Author Response · Authors · 2025-11-28
>
> We sincerely thank the reviewer for the positive assessment of our paper, the clear summary, and the thoughtful feedback. We especially appreciate the recognition of the theoretical motivation, the practical relevance of HRQ, and the consistent improvements across both supervised and unsupervised settings. We also thank the reviewer for the high overall score and the recommendation for acceptance.
>
> ### W1:
>
> We agree that HRQ is designed for settings where the data exhibit **latent hierarchical structure**. However, we note that such structure is far more common than it may initially appear. In practice, **any graph-structured domain contains at least one hierarchy** through its spanning trees, and many real-world datasets: social graphs, taxonomies, product catalogs, ontologies, dependency graphs, recommendation datasets. They all implicitly encode hierarchical organization even if they are not explicitly labeled as such.
>
> Nonetheless, we fully acknowledge that HRQ’s inductive bias is most beneficial when a clear or moderately strong hierarchy is present. In domains where the structure is very flat or unorganized, we expect smaller or no benefit from HRQ. This is important area for future direction. However, investigating it requires a scruputous investigation across different dataset. This direction might also lead to some hybrid models that combine euclidean and hyperbolic geometry for residual quantization. At the moment, a simple solution we can think of is simply trying both HRQ-VAE and RQ-VAE and choosing whichever works better on the validation set.
>
> ### W2:
>
> We agree that HRQ-VAE inherits several well-known limitations from **hyperbolic neural networks**. These include the frequent necessity of training in **float64** to maintain numerical stability, as well as conceptual and implementation challenges that arise when extending hyperbolic models to more complex architectures. In practice, deeper or more expressive encoders can exhibit **training instabilities.** We have also **added a clear note to the manuscript highlighting the float64 requirement** and its implications for training efficiency.
>
> There is active research aimed at mitigating these issues, but these solutions are still emerging and often require substantial engineering effort to integrate reliably. Because of these challenges, we deliberately limited HRQ-VAE to relatively simple encoders and did not explore more demanding architectures such as convolutional hyperbolic models. We consider this an **exciting direction for future work**, and we believe that advances in numerically stable hyperbolic training will make such extensions increasingly feasible.
>
> Once again, we would like to express our sincere gratitude to the reviewer for the exceptionally positive assessment. We are delighted that the core strengths of the paper: its strong theoretical motivation, the novelty of integrating hyperbolic geometry into residual quantization, the clarity of the presentation, and the consistent empirical improvements, all resonated so clearly.

---

### Meta-Review · Area_Chair_3So3 · 2026-01-06

**Summary:**

This paper looks at the problem of learning quantized embeddings of data in hyperbolic space and then using these embedding for extracting hierarchies.

The paper is initially leaning towards reject initially. The initial scores are 8 (cMFE), 4 (brqf), 2 (rJFw), and 2 (6yTa).

The main concerns of the reviewers are as follows:

1. Lack of comparison with prior work (reviewers brqf, rJFw, 6yTa). I think this is primary major concern of the reviewers. That the paper has not sufficiently demonstrated the advantages of their method.

2. Issues with using Euclidean distance versus hyperbolic distance (6yTa)

3. Down stream applications and analysis of hierarchies discovered (6yTa)

**Reviewer Concerns:**

## Addressed concerns

I think the authors do a particularly convincing job of addressing the concern about downstream applications and analysis of the hierarchies discovered. They added multiple experiments and figures.

## Only Partially Addressed

However the other two concerns of comparisons to baselines is only partially addressed. The rebuttal argues that certain baselines are not directly comparable or lack implementations, which is helpful context. However, the response does not fully resolve the core concern: the paper would be significantly stronger with direct comparisons and/or carefully chosen proxy baselines.

For example just a hyperbolic encoder, no quantization. How does the reconstruction error change when doing this. Suppose the code book is fixed and not trained or another static quantization technique is used.

Additionally, the answer to using the Euclidean distance instead of hyperbolic distance is not convincing at all. In particular, I do not see why codebook vectors are close to the residuals, they could be quite far away. In particular, it is not clear why the relevant geometry should be locally Euclidean in the regime that matters for performance, nor is it supported by a targeted ablation demonstrating that a fully hyperbolic alternative is either unstable or worse (and why). This remains a significant methodological concern.

**Reviewer Scores:**

cMFE (8): Likely no change (stays 8). This reviewer was already strongly positive and the rebuttal does not introduce new weaknesses.

brqf (4): Likely +2 (4 to 6). The rebuttal improves the story and addresses some questions, but the missing baseline comparisons likely remain a blocker for a strong score increase.

rJFw (2): Likely +2 (2 to 4). The rebuttal may move them somewhat, but the core “not sufficiently demonstrated vs alternatives” concern plausibly remains.

6yTa (2): Unlikely to change. While the authors address the downstream/hierarchy evidence better, the methodological concern (hyperbolic vs Euclidean components) and baseline gaps likely keep this reviewer in reject territory.

---

### Decision · Program_Chairs · 2026-01-26

Reject